# Meta-Diffu$B$: A Contextualized Sequence-to-Sequence Text Diffusion Model with Meta-Exploration

**Yun-Yen Chuang**[1,2]**, Hung-Min Hsu**[3]**, Kevin Lin**[4]**,**
**Chen-Sheng Gu**[1,2]**, Ling Zhen Li**[1,2]**, Ray-I Chang**[2]**, Hung-yi Lee**[2]
[1]Maxora AI  [2]National Taiwan University
[3]University of Washington  [4]Microsoft
yunyenchuang@maxora.ai, hmhsu@uw.edu, keli@microsoft.com,
chenshenggu@maxora.ai, lingzhenli@maxora.ai,
rayichang@ntu.edu.tw, hungyilee@ntu.edu.tw

## Abstract

The diffusion model, a new generative modeling paradigm, has achieved significant success in generating images, audio, video, and text. It has been adapted for sequence-to-sequence text generation (Seq2Seq) through DiffuSeq, termed the S2S-Diffusion model. Existing S2S-Diffusion models predominantly rely on fixed or hand-crafted rules to schedule noise during the diffusion and denoising processes. However, these models are limited by non-contextualized noise, which fails to fully consider the characteristics of Seq2Seq tasks. In this paper, we propose the Meta-Diffu$B$ framework—a novel scheduler-exploiter S2S-Diffusion paradigm designed to overcome the limitations of existing S2S-Diffusion models. We employ Meta-Exploration to train an additional scheduler model dedicated to scheduling contextualized noise for each sentence. Our exploiter model, an S2S-Diffusion model, leverages the noise scheduled by our scheduler model for updating and generation. Meta-Diffu$B$ achieves state-of-the-art performance compared to previous S2S-Diffusion models and fine-tuned pre-trained language models (PLMs) across four Seq2Seq benchmark datasets. We further investigate and visualize the impact of Meta-Diffu$B$'s noise scheduling on the generation of sentences with varying difficulties. Additionally, our scheduler model can function as a "plug-and-play" model to enhance DiffuSeq without the need for fine-tuning during the inference stage. [1]

## 1 Introduction

The diffusion model, a novel generative approach, operates through a two-step process: it first introduces noise to real data and then systematically removes this noise to facilitate data generation [12, 40, 30]. This model has demonstrated significant efficacy across several domains, including image [13, 29, 38], audio [36, 18], video [18, 14], and text generation [1, 15, 21, 3, 24, 34]. The diffusion model utilizes a technique known as noise scheduling to control the amount of noise imposed at each diffusion step [12]. DiffuSeq [8] has adapted this model to discrete generation tasks like sequence-to-sequence text generation (Seq2Seq), under a framework termed S2S-Diffusion. However, DiffuSeq employs fixed noise scheduling and does not accommodate the specific characteristics of Seq2Seq tasks [45, 44].

Seq2Seq is a foundational technique in natural language processing (NLP) that generates target sentences from specified conditional sentences. It supports a range of downstream tasks, including

---

[1]Code and datasets for Meta-Diffu$B$ are available at: `https://github.com/Meta-DiffuB/Meta-DiffuB`.

38th Conference on Neural Information Processing Systems (NeurIPS 2024).

language translation [41], image captioning [35], conversational modeling [39], and text summarization [28]. For Seq2Seq tasks, it is more reasonable to impose different levels of noise to each sentence in S2S-Diffusion models to address the varying semantic and contextual difficulties of generating sentences. This noise scheduling strategy can better adapt to the semantic characteristics and generation difficulties of each sentence, thereby improving the model's performance in various generation tasks. To meet the unique demands of S2S Diffusion, we introduce a contextualized noise-scheduling strategy that accounts for the semantics of each conditional sentence and adapts to different training epochs. Existing S2S-Diffusion models, such as DiffuSeq, lack flexibility due to their reliance on fixed, non-contextualized noise-scheduling strategies. Furthermore, models like SeqDiffuSeq [45] and Dinoiser [44], which propose adaptive noise scheduling, are also limited by their non-contextualized approach.

To address the semantics of discrete conditional sentences for contextualized noise scheduling, we introduce a novel scheduler-exploiter framework, Meta-Diffu$B$, which achieves trainable noise-scheduling inspired by Meta-Exploration [43]. Within this framework, our scheduler model dynamically schedules noise to train our exploiter model, which is updated based on the performance rewards it generates. Our exploiter model, an S2S-Diffusion model, leverages the noise scheduled by the scheduler model for updates and generation. By design, Meta-Diffu$B$ naturally implements contextualized noise scheduling. It achieves state-of-the-art performance on four Seq2Seq benchmark datasets, outperforming existing S2S-Diffusion models [8, 45, 44] and fine-tuned pre-trained language models (PLMs) [10, 33].

In summary, we make three primary contributions with Meta-Diffu$B$:

- We introduce and demonstrate the application of Meta-Exploration to diffusion models in Section 3, proposing Meta-Diffu$B$ as a strategy to enhance S2S-Diffusion models. Our main results, presented in Section 6.1, confirm that Meta-Diffu$B$ achieves state-of-the-art performance across four benchmark datasets.

- We detail the operation of our scheduler model in Section 6.2, highlighting its capability to schedule noise. The noise scheduling approach of our scheduler model—applying less noise to the harder sentences and more to the easier ones—enhances the diversity and quality of the generated text.

- We reveal that our scheduler model can function as a "plug-and-play" model, easily integrated into existing S2S-Diffusion models to enhance inference performance, as detailed in Section 6.3.

## 2   Problem Statement, Preliminary

### 2.1   Problem Statement

In this work, we focus on sequence-to-sequence text generation tasks. Given a conditioning sentence of length $m$, $\mathbf{w}^x = \{w_1^x, \ldots, w_m^x\}$, our objective is to train a diffusion model capable of generating a target sentence of length $n$, $\mathbf{w}^y = \{w_1^y, \ldots, w_n^y\}$, based on the conditional sentence. Here, $\mathbf{w}^x$ and $\mathbf{w}^y$ represent the conditional and target sentences, respectively.

### 2.2   Preliminary

DiffuSeq [8] primarily follows the transformation method of Diffusion-LM [21] and incorporates the diffusion and denoising processes from [12]. In the diffusion process, Diffusion-LM transforms discrete sentences into a continuous space. Given the real-world training sentence pair $\mathbf{w}^{x \oplus y}$, concatenated by $\mathbf{w}^x$ and $\mathbf{w}^y$, Diffusion-LM uses an embedding function $\text{emb}$ to transform $\mathbf{w}^{x \oplus y}$ into continuous space, thereby obtaining the distribution $\mathbf{z}_0 \sim q(\mathbf{z})$, where $q$ represents the diffusion process. Then, $\mathbf{z}_0$ is subjected to imposed noise, diffusing into a standard Gaussian distribution $\mathbf{z}_T \sim \mathcal{N}(0, \mathbf{I})$. At each diffusion step $t \in [1, 2, \ldots, T]$, the noise is regulated by $q(\mathbf{z}_t | \mathbf{z}_{t-1}) = \mathcal{N}(\mathbf{z}_t; \sqrt{1 - \beta_t} \mathbf{z}_{t-1}, \beta_t \mathbf{I})$, where $\beta_t \in (0, 1)$ controls the amount of noise imposed at each diffusion step. We denote $\boldsymbol{\beta}$ as containing a set of noise values $\beta_t$, where a larger $\beta_t$ indicates more Gaussian noise imposed at that diffusion step. When $t$ is large enough, $\mathbf{z}_0$ gradually evolves into a standard Gaussian noise distribution. The random distribution is gradually reduced in noise during the denoising process to regenerate target sentences. The denoising process, which recovers $\mathbf{z}_0$ by

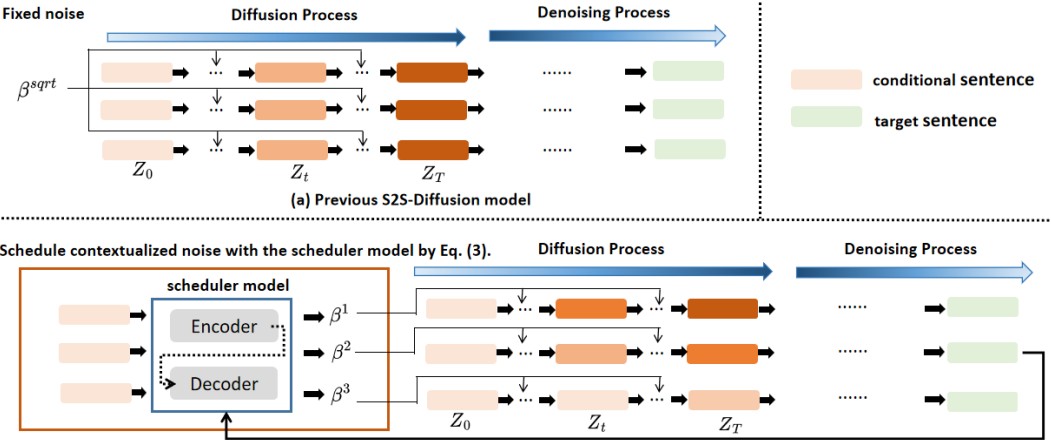

Figure 1: Comparison between S2S-Diffusion model (*i.e.*, DiffuSeq [21]) and the proposed Meta-Diffu$B$. The shades of color represent different amounts of noise being imposed. Different from prior works that use a fixed noise, we introduce a novel scheduler-exploiter framework, Meta-Diffu$B$, which achieves trainable noise scheduling inspired by Meta Exploration. Our scheduler model schedules contextualized noise, enhancing the training and generation of the S2S-Diffusion model, resulting in state-of-the-art (SOTA) performance compared to previous S2S-Diffusion models, as detailed in Section 4.

reducing the noise in $\mathbf{z}_t$, can be defined as follows:

$$p_\theta(\mathbf{z}_{0:T}) = p(\mathbf{z}_T) \prod_{t=1}^{T} p_\theta(\mathbf{z}_{t-1}|\mathbf{z}_t). \tag{1}$$

Diffusion-LM employs a trained, parameterized denoising distribution $\mathbf{z}_{t-1} \sim p_\theta(\mathbf{z}_{t-1}|\mathbf{z}_t)$ to gradually recover $\mathbf{z}_t$ from noise. This denoising distribution, parameterized by $\theta$, is tailored to fit the posterior distribution $q(\mathbf{z}_{t-1}|\mathbf{z}_t, \mathbf{z}_0)$ of the forward process. The key difference between DiffuSeq [8] and Diffusion-LM [21] is that DiffuSeq imposes noise only on the target sentence part of $\mathbf{z}_t$ to achieve classifier-free S2S Diffusion, termed Partial Noise [8]. Due to the implementation of Partial Noise in the diffusion process, conditional denoising is inherently classifier-free. To transform the continuous $\mathbf{z}_0$ target sentences back into discrete sentences $\mathbf{w}^y$, previous S2S-Diffusion models use a Rounding Operation [21] to map the target sentence part of $\mathbf{z}_0$ into $\mathbf{w}^y$. The Rounding Operation is a method for choosing the most probable word for each position [21]. The denoising process primarily utilizes the variational lower bound ($\mathcal{L}_{\text{VLB}}$) to optimize the negative log-likelihood [12]. Through the simplification and derivation from DiffuSeq [8], the training objective function for S2S-Diffusion models can be defined as:

$$\min_\theta \mathcal{L}_{\text{VLB}} = \min_\theta [\sum_{t=2}^{T} \|\mathbf{z}_0 - f_\theta(\mathbf{z}_t, t)\|^2 + \|\operatorname{emb}(\mathbf{w}^{x \oplus y}) - f_\theta(\mathbf{z}_1, 1)\|^2 + \mathcal{R}(\|\mathbf{z}_0\|^2)], \tag{2}$$

where learning process $p_\theta(\mathbf{z}_{t-1}|\mathbf{z}_t)$ is modeled as Transformer model $f_\theta$. Previous diffusion models deploy $\boldsymbol{\beta}$ by dividing the interval between the minimum value $\beta_1$ and the maximum value $\beta_T$ using a mathematical function to determine the fixed noise sequence $\{\beta_1, ..., \beta_T\} \in \boldsymbol{\beta}$, as described in [12]. The mathematical function used by Diffusion-LM and DiffuSeq [21] is the $sqrt$ function, which has demonstrated superior performance in text generation compared to other fixed mathematical functions. However, DiffuSeq's noise scheduling is constrained by its non-contextual approach; it does not account for the semantics of each conditional sentence nor does it adapt to different training epochs.

# 3 Methodology

In this work, we propose a scheduler-exploiter framework named Meta-Diffu$B$ for training S2S-Diffusion models with contextualized noise. Inspired by [43], our Meta-Diffu$B$ includes a scheduler model, $B_\psi$, parameterized by $\psi$, and an exploiter model, $D_\theta$, parameterized by $\theta$. $B_\psi$, a simple Seq2Seq model, considers the semantics of conditional sentences to schedule contextualized $\boldsymbol{\beta}$ for updating $D_\theta$ and is also updated based on the learning effectiveness of $D_\theta$—which refers to how well the exploiter learns. Our exploiter, $D_\theta$, an S2S-Diffusion model, leverages the noise scheduled by $B_\psi$ for its updating and generation. The framework of our Meta-Diffu$B$, compared with DiffuSeq, is visualized in Figure 1.

## 3.1 Noise Scheduling in the Scheduler Model

In this work, we propose a simple two-step approach for our scheduler $B_\psi$, which is a Seq2Seq model, to schedule $\boldsymbol{\beta}$—a set of noise values $\beta_t$. Here, a larger $\beta_t$ indicates more noise imposed on the data. The input to $B_\psi$ is consistently $\mathbf{w}^x$ across both training and inference stages. Instead of directly scheduling the values of $\boldsymbol{\beta}$, $B_\psi$ outputs a series of Meta-Instructions, simplifying the training into a time-series binary classification problem. In the first step, $B_\psi$ samples a series of Meta-Instructions $\iota^x = \{\iota_1, \ldots, \iota_t, \ldots, \iota_T\}$ from $\mathbf{w}^x$, where each $\iota_t$ is labeled either True or False. We propose a 'skipping' method: a True label directs $B_\psi$ to increase the noise by selecting $\beta_{t+1}$ for the next diffusion step, whereas a False label maintains the same noise level $\beta_t$. In the second step, we transform $\iota^x$ using the fixed noise $sqrt$-function $\boldsymbol{\beta}^{sqrt} = \{\beta_1, \ldots, \beta_T\}$, as deployed by [21, 8], through the 'skipping' method to generate the new noise values $\boldsymbol{\beta}^x = \{\beta_1^x, \ldots, \beta_T^x\}$. For example, with the continuous Meta-Instructions $\iota^x = \{T, F, T\}$ and fixed noise values $\{1, 2, 3\}$, our new scheduling of noise values will be $\{1, 1, 2\}$. If consecutive scheduling noise values are the same, no additional noise is introduced at that diffusion step [12]. Our two-step approach maintains the same diffusion steps for parallel operations and contextualized $\boldsymbol{\beta}^x$ in the diffusion process. We utilize a Policy Gradient to update our scheduler model following Meta-Exploration, addressing the non-differentiability of our two-step approach. The noise-scheduling mechanism of our scheduler model can be defined by the following equations:

$$\begin{aligned} \iota^x &= B_\psi(\mathbf{w}^x) \\ \boldsymbol{\beta}^x &= skipping(\iota^x, \boldsymbol{\beta}^{sqrt}). \end{aligned} \tag{3}$$

## 3.2 Training the Exploiter

Unlike previous S2S-Diffusion models [8, 45, 44] that employ fixed or hand-crafted noise scheduling, we utilize contextualized $\boldsymbol{\beta}^x$ to impose noise during the diffusion process. We also implement Partial Noise to achieve classifier-free S2S Diffusion [8]. In the denoising process, our exploiter model $D_\theta$ restores the diffused data to generate the target sentences. During the diffusion process, we adopt the transformation method of Diffusion-LM to obtain $\text{emb}(\mathbf{w}^{x \oplus y})$, as described in Section 2. We extend the original diffusion chain to a new Markov transition with our $\boldsymbol{\beta}^x$: $q_\phi(\mathbf{z}_0|\mathbf{w}^{x \oplus y}) = \mathcal{N}(\text{emb}(\mathbf{w}^{x \oplus y}), \beta_0^x \mathbf{I})$ [21, 8]. Consequently, we can implement the objective function indicated in Section 2, derived from previous classifier-free S2S-Diffusion methods, to update our exploiter model $D_\theta$ [8]. The training objective function for our exploiter model $D_\theta$ in collaboration with $B_\psi$ can be defined as follows:

$$\nabla_\theta J(\theta) = \min_\theta [\sum_{t=2}^{T} \|\mathbf{z}_0 - \nabla_\theta D_\theta(\mathbf{z}_t^{B_\psi}, t)\|^2 + \|\text{emb}(\mathbf{w}^{x \oplus y}) - \nabla_\theta D_\theta(\mathbf{z}_1^{B_\psi}, 1)\|^2 + \mathcal{R}(\|\mathbf{z}_0\|^2)]. \tag{4}$$

Since $\mathbf{z}_0$ is not diffused, there is no need to add the superscript of $B_\psi$. $J(\theta)$ is denoted as the gradient for updating exploiter model $D_\theta$. Then, we can update our exploiter model $D_\theta$'s network weights:

$$\theta' \rightarrow \theta + \nabla_\theta J(\theta). \tag{5}$$

## 3.3 Contextualized Inference with Meta-Diffu$B$

In the inference stage, if our goal is to generate outputs based on $\mathbf{w}^x$, $B_\psi$ predicts contextualized $\boldsymbol{\beta}^x$ using $\mathbf{w}^x$, as demonstrated in Section 3.1. We then concatenate $\text{emb}(\mathbf{w}^x)$—transformed from

**Algorithm 1** Meta-Diffu$B$

---

**Require:** exploiter model $D_\theta$; scheduler model $B_\psi$; conditional sentences $\mathbf{w}^x$ and target sequences $\mathbf{w}^y$ from dataset.

  Initialize exploiter model $D_\theta$, scheduler model $B_\psi$ with random weights $\theta$ and $\psi$.
  **repeat**
    **for** $e$ in $1 : \mathcal{E}$ exploration epochs **do**
      $B_\psi$ schedules noise $\boldsymbol{\beta}^x$ by Eq. (3).
      $\theta^e \leftarrow \theta + \nabla_\psi J(\theta)^e$ by Eq. (4) and Eq. (5)
      Estimate the Meta-Reward $\mathcal{R}^e_{B_\psi}$ as described in Section 3.4.
      Compute the gradient $\nabla J(\psi)^e$ by Eq. (6).
    **end for**
    $\psi' \leftarrow \psi + \sum_{e=1}^{\mathcal{E}} \nabla J(\psi)^e$ by Eq. (7). { Scheduler Update }
    $B_{\psi'}$ schedules noise $\boldsymbol{\beta}^x$ by Eq. (3).
    $\theta' \leftarrow \theta + \nabla_\theta J(\theta)'$ by Eq. (4) and Eq. (5). { Exploiter Update }
  **until** Meta-Diffu$B$ converges

---

$\mathbf{w}^x$—with a randomly sampled $\mathbf{y}_T \sim \mathcal{N}(0, I)$ to form $\mathbf{z}_T$. Our $D_\theta$ predicts $\mathbf{z}_0$ directly from $\mathbf{z}_t$ and uses $\boldsymbol{\beta}^x$ to convert the predicted $\mathbf{z}_0$ into $\mathbf{z}_{t-1}$. This step-by-step denoising process progressively recovers $\mathbf{z}_t$ back to $\mathbf{z}_0$, following the methodologies outlined in [21, 45, 44]. Finally, we use a Rounding Operation to convert the target sentence part of $\mathbf{z}_0$ into discrete target sentences.

### 3.4 Estimating the Meta-Reward of the Scheduler Model

In this section, we estimate the Meta-Reward of our scheduler model, which reflects the learning effectiveness of $D_\theta$. We let $D_\theta$ generate $\mathbf{Y}_{D_\theta}$ and $D_{\theta'}$ generate $\mathbf{Y}_{D_{\theta'}}$, respectively, where $\mathbf{Y}$ denotes the generated $\mathbf{w}^y$ [21]. We assess the rewards for $\mathbf{Y}_{D_\theta}$ and $\mathbf{Y}_{D'_{\theta'}}$, denoted as $R_{D_\theta}$ and $R_{D_{\theta'}}$ respectively, which represent the rewards for $D_\theta$ and $D_{\theta'}$. In this study, we utilize the BLEU score to quantify these rewards. Consequently, the reward for the scheduler model (i.e., Meta-Reward) is defined as $R_{B_\psi} = R_{D_{\theta'}} - R_{D_\theta}$.

### 3.5 Training the Scheduler Model with Meta-Reward

Since $B_\psi$ generates Meta-Instructions to diffuse sentences using our two-step approach described in Section 3.1, we update the scheduler via policy gradients, incorporating both Meta-Instructions and the calculated Meta-Rewards [43]. The training objective function for our scheduler model is defined as follows:

$$\nabla_\psi J(\psi) = \sum_{t=1}^{T} \nabla_\psi B_\psi(\iota_t^x \mid \mathbf{w}^x) \cdot R_{B\psi}. \tag{6}$$

After we obtain $\nabla_\psi J(\psi)$, we can update $B_\psi$'s network weights:

$$\psi' = \psi + \nabla_\psi J(\psi). \tag{7}$$

### 3.6 Exploration Epochs

Inspired by the exploration epochs of Meta-Exploration [43, 5, 19], we iteratively execute the procedures from Section 3.1 to Section 3.4 to collect various indicators of learning effectiveness from $D_\theta$ for updating $B_\psi$. In practice, we keep the network weights of $D_\theta$ fixed until the exploration epochs are completed. This approach ensures that the scheduler model schedules noise to $D_\theta$ with consistent network weights, promoting stable training [5]. Additionally, we can conduct the exploration epochs in parallel to save time by collecting learning effectiveness from $D_\theta$ under consistent network weights. After accumulating the gradients for $B_\psi$ from these exploration epochs, we update $B_\psi$ to $B_{\psi'}$, which in turn schedules new noise to update $D_\theta$ to $D_{\theta'}$. In summary, we present Algorithm (1) to detail the full training process of the proposed Meta-Diffu$B$. The number of exploration epochs is denoted by $\mathcal{E}$, with $e \in \{1, ..., \mathcal{E}\}$ indexing the exploration epochs.

# 4 Experiments

In this section, we conduct experiments to verify the performance of our Meta-Diffu$B$ on four benchmark Seq2Seq datasets [48, 6, 17, 8]. We benchmark Meta-Diffu$B$ against previous S2S-Diffusion models and fine-tuned pre-trained language models (PLMs), using the same datasets and training settings as employed by DiffuSeq [8].

## 4.1 Datasets

In our experiment, we use four datasets: the Commonsense Conversation dataset (CC) [48], the Quasar-T dataset (QT) [6], the Wiki-Auto dataset (WA) [17], and the Quora Question Pairs dataset (QQP) [8]. These datasets consider a variety of tasks, including open-domain dialogue generation, question generation, text simplification, and paraphrase generation tasks, all within Seq2Seq contexts. For a fair comparison, we employ the same datasets with identical settings for training all mentioned models, as outlined in [8, 45]. Detailed settings of these datasets are provided in Appendix A.

## 4.2 Baselines

We compare the proposed Meta-Diffu$B$ with previous S2S-Diffusion models, including DiffuSeq [8], Dinoiser [44], and SeqDiffuSeq [45]. DiffuSeq employs a fixed noise pattern in the training and inference stages using a $sqrt$ function and has been successfully introduced to the Seq2Seq task as the basic diffusion model. We also compare Meta-Diffu$B$ with Dinoiser and SeqDiffuSeq, which are existing S2S-Diffusion models that focus on noise scheduling. Dinoiser and SeqDiffuSeq utilize hand-crafted rules that provide adaptive but not contextualized noise scheduling. Additionally, following [8, 45], we compare our Meta-Diffu$B$ with three PLMs on Seq2Seq tasks. These PLMs include the fine-tuned GPT-2-base (GPT2-base) [33], fine-tuned GPT-2-large (GPT2-large), and fine-tuned Levenshtein Transformer (LevT) [10]. We detail these baselines in Appendix B.

## 4.3 Training Setting

Our exploiter model employs the same network architecture and settings as DiffuSeq [8]. Our scheduler model uses the same network architecture as described in [4]. The exploiter model and scheduler architectures are detailed in Appendix C. For consistent comparison, all S2S-Diffusion models [8, 44, 45] follow the experimental settings of prior research [8] and are trained from scratch. The diffusion step count is set at 2,000, and the maximum sequence length is 128. The Minimum Bayes risk (MBR) [23] decoding size, denoted as $|S|$, is 10; this involves generating sentences from 10 random seeds and selecting the best output sequence. Details on the implementation of MBR for all S2S-Diffusion models can be found in Appendix 6. The total batch size for both training and testing phases is 2048. Experiments are conducted on NVIDIA A100 Tensor Core GPUs, utilizing 4 GPUs for training and a single GPU for inference.

### 4.3.1 Discussion of Computational Intensity

To ensure a fair comparison during parallel exploration epochs, we avoid increasing the total batch size. Instead, we reduce the batch size by dividing the total batch size by the number of exploration epochs deployed. In this work, we set the number of exploration epochs to 32 and the batch size to 64. To update our scheduler, we run parallel exploration epochs every 100 training epochs with a total batch size of 2048. The increased computational complexity of applying Meta-Diffu$B$ to DiffuSeq is presented in Table 1.

Table 1: Computational complexity increase when applying Meta-Diffu$B$ to DiffuSeq.

| Method | Increased Parameters (%) | Increased Training Time (%) | Increased Inference Time (%) |
|---|---|---|---|
| Meta-Diffu$B$ | 2.2% | 5% | 0.5% |

## 4.4 Evaluation Metrics

To ensure a fair comparison, we follow the same evaluation metric settings as those used in previous S2S-Diffusion models [8, 45]. For quality assessment, we utilize standard text generation metrics such

as BLEU [27], ROUGE-L [2], and BERTScore [46], where higher scores indicate better performance. For diversity assessment, we apply general text generation diversity metrics, including Distinct Unigram (Dist-1)[2] and Self-BLEU[27], where lower scores of Self-BLEU and higher scores of Dist-1 signify better performance. Due to the application of multiple evaluation metrics (such as BLEU, ROUGE-L, BERTScore, Dist-1, and Self-BLEU), we also use Mean-Rank (M-R) to measure whether each model performs the best across multiple metrics [20]. A lower Mean-Rank score indicates consistently better performance across various metrics in the dataset. Details on the evaluation metric settings and their explanations are provided in Appendix D.

## 5 Model-Agnostic Characteristics of Meta-Diffu$B$

We conduct experiments on applying our Meta-Diffu$B$ to other S2S-Diffusion models. Specifically, we use Meta-Diffu$B$ to modify the handcrafted noise-scheduling strategies of Dinoiser [44] and SeqDiffuSeq [45] on the WA and QQP datasets. The results, shown in Table 2, demonstrate that Meta-Diffu$B$ can be considered a model-agnostic method for enhancing the performance of other S2S-Diffusion models. Additionally, we provide results for applying our Meta-Diffu$B$ to RDM [47] (based on D3PM [47]) and other recent S2S-Diffusion models [42, 7, 22, 9], which are based on DiffuSeq [8] on machine translation datasets [31, 26] in Appendix E.

Table 2: Results of applying our Meta-Diffu$B$ ($D_\theta$ = a specific S2S-Diffusion model) to other S2S-Diffusion models [8, 45, 44]. The specific S2S-Diffusion model used in the exploiter model is indicated by the assignment of $D_\theta$. Outcomes where Meta-Diffu$B$ outperforms previous S2S-Diffusion models are highlighted in **bold**. A star ($\star$) indicates results reported directly from previous studies, while a dagger ($\dagger$) signifies that we reproduced the results because the original studies did not report them using the same metrics on these datasets.

| Tasks | Methods | BLEU ($\uparrow$) | BERTScore ($\uparrow$) | Dist-1 ($\uparrow$) |
|---|---|---|---|---|
| QQP | $\star$ DiffuSeq | 0.2413 | 0.8365 | 0.9807 |
| | Meta-Diffu$B$ ($D_\theta$ = DiffuSeq) | **0.2552** | **0.8821** | **0.9922** |
| | $\star$ SeqDiffuSeq | 0.2434 | 0.8400 | 0.9807 |
| | Meta-Diffu$B$ ($D_\theta$ = SeqDiffuSeq) | **0.2632** | **0.8919** | **0.9902** |
| | $\dagger$ Dinoiser | 0.1949 | 0.8036 | 0.9723 |
| | Meta-Diffu$B$ ($D_\theta$ = Dinoiser) | **0.2271** | **0.8525** | **0.9752** |
| WA | $\star$ DiffuSeq | 0.3622 | 0.8126 | 0.9264 |
| | Meta-Diffu$B$ ($D_\theta$ = DiffuSeq) | **0.3877** | **0.8233** | **0.9355** |
| | $\star$ SeqDiffuSeq | 0.3712 | 0.8214 | 0.9077 |
| | Meta-Diffu$B$ ($D_\theta$ = SeqDiffuSeq) | **0.3957** | **0.8451** | **0.9412** |
| | $\dagger$ Dinoiser | 0.2388 | 0.6787 | 0.8421 |
| | Meta-Diffu$B$ ($D_\theta$ = Dinoiser) | **0.2471** | **0.7285** | **0.8694** |

## 6 Experiments of Minimum Bayes Risk Decoding

Diffusion-LM proposes using Minimum Bayes Risk (MBR) to improve generation. Following the methods described in [45, 8], we allow all S2S-Diffusion models to generate a set of candidate sentences from 10 random seeds and select the best output sequence that achieves the minimum expected risk under a meaningful loss function. Specifically, in this work, we employ the BLEU score as our loss function to evaluate performance, following the approach used in DiffuSeq [8]. We compare our Meta-Diffu$B$ ($D_\theta$ = DiffuSeq) with DiffuSeq [8] and GPT-2 [33], using MBR decoding [21, 8, 45] on the WA and QQP datasets as described in DiffuSeq [8]. We specifically select GPT2-large and GPT2-base for comparison based on their superior performance on these datasets [8]. In this experiment, we apply MBR decoding to all three models while gradually increasing the candidate sentence size $|S|$. The results of the MBR decoding are presented in Figure 2.

Figure 2 shows that our Meta-Diffu$B$ ($D_\theta$ = DiffuSeq) can generate a more diverse array of candidate sentences, achieving better results as the candidate size $|S|$ increases. The diversity of these candidate sentences determines the upper bound of MBR performance [21, 8]. Our Meta-Diffu$B$ ($D_\theta$ = DiffuSeq) consistently outperforms both GPT2-base and DiffuSeq across all candidate size settings

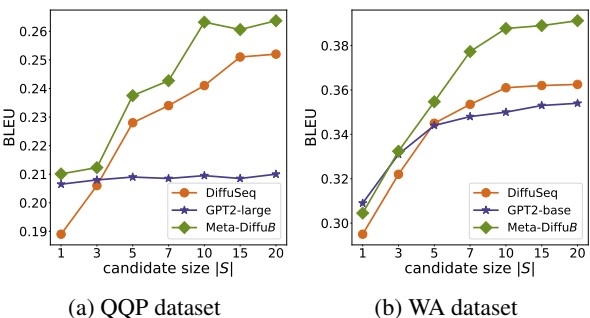

(a) QQP dataset      (b) WA dataset

Figure 2: Increase in BLEU score with varying candidate sizes $|S|$ on the QQP and WA datasets.

on the QQP dataset. As the candidate size grows, Meta-Diffu$B$ ($D_\theta$ = DiffuSeq) also surpasses GPT2-base on the WA dataset. Notably, Meta-Diffu$B$ ($D_\theta$ = DiffuSeq) exhibits diverse generation capabilities and achieves significantly better performance than DiffuSeq in MBR experiments.

## 6.1 Experiment with Seq2Seq Benchmark Datasets

We demonstrate the performance of our Meta-Diffu$B$ ($D_\theta$ = DiffuSeq) in Table 3. This table shows that Meta-Diffu$B$ ($D_\theta$ = DiffuSeq) outperforms other PLMs [33, 10] and S2S-Diffusion models [8, 44, 45] in terms of generation quality and diversity, achieving the lowest M-R scores across four datasets. Moreover, Meta-Diffu$B$ ($D_\theta$ = DiffuSeq) demonstrates significant improvements over previous S2S-Diffusion models [8, 45, 44] on all evaluation metrics considered.

## 6.2 Contextualized Noise Scheduling of Meta-Diffu$B$

For the experiment involving contextualized noise of Meta-Diffu$B$ ($D_\theta$ = DiffuSeq), we selected the 200 hardest and 200 easiest generated sentences, labeled as (H) and (E), respectively. All models listed in Table 3 assessed the generation difficulty of each sentence using BLEU scores, with lower BLEU indicating higher difficulty. The performance of generating (H) and (E) is evaluated in terms of BLEU and Self-BLEU, as shown in Table 4. We also detail the results of Table 4 evaluated by other metrics in Appendix G. We tasked all S2S-Diffusion models with scheduling noise for sentences (H) and (E), as shown in Figure 3. Since our Meta-Diffu$B$ ($D_\theta$ = DiffuSeq) assigns specific noise to each sentence, we averaged the noise values for clearer visualization. Figure 3 displays the last 10 diffusion steps, as the noise differences in the initial steps are minimal. In Table 4, Meta-Diffu$B$ ($D_\theta$ = DiffuSeq) consistently outperforms other S2S-Diffusion models [44, 45, 8] in terms of generation quality and diversity for sentences (E) and (H). Notably, when generating the more challenging sentences (H), Meta-Diffu$B$ Meta-Diffu$B$ ($D_\theta$ = DiffuSeq) maintains its performance, whereas other S2S-Diffusion models [44, 45, 8] experience a decline in both quality and diversity. Figure 3 shows the benefits of Meta-Diffu$B$ ($D_\theta$ = DiffuSeq), which strategically imposes different noise levels for sentences (E) and (H). This noise-scheduling approach—applying less noise to the harder sentences (H) and more to the easier sentences (E)—enhances the diversity and quality of the generated text. The superior example of Meta-Diffu$B$ in generating the hardest sentences (H) is further showcased in Appendix F, demonstrating its performance relative to other S2S-Diffusion models [44, 45, 8]. We also provide a more detailed discussion about the noise strategy of our Meta-Diffu$B$ in Appendix H.

## 6.3 Plug-and-Play Experiments with the Scheduler Model

Our pre-trained scheduler model, trained under Meta-Diffu$B$ ($D_\theta$ = DiffuSeq), which incorporates the semantics of discrete sentences, demonstrates its effectiveness by scheduling noise for pre-trained DiffuSeq [8] models across various datasets. The results, presented in Table 5, show that our pre-trained scheduler model, when applied across different datasets, enhances the performance of pre-trained DiffuSeq models without any fine-tuning during the inference stage. This confirms that our scheduler model can function as a plug-and-play model across these datasets. Additionally, we provide further experiments on different pre-trained schedulers under various S2S-Diffusion settings, as well as results on additional datasets in Appendix I.

Table 3: We present the results of our Meta-Diffu$B$ ($D_\theta$ = DiffuSeq) compared with other models across four Seq2Seq datasets. We report the scores of DiffuSeq and PLMs from [8]. A star ($\star$) indicates results reported directly from previous studies, while a dagger ($\dagger$) signifies that we reproduced the results because the previous studies did not report them using the same metrics on these datasets. The best results among S2S-Diffusion models are underlined, and the overall best results are in **bold**.

| Tasks | Methods | BLEU ($\uparrow$) | ROUGH-L ($\uparrow$) | BERTScore ($\uparrow$) | Dist-1 ($\uparrow$) | Self-BLEU ($\downarrow$) | M-R ($\downarrow$) |
|---|---|---|---|---|---|---|---|
| QQP | $\star$ GPT2-base | 0.1980 | 0.5212 | 0.8246 | 0.9798 | 0.5480 | 5.20 |
| | $\star$ GPT2-large | 0.2059 | 0.5415 | 0.8363 | 0.9819 | 0.7325 | 3.80 |
| | $\star$ LevT | 0.2268 | 0.5795 | 0.8344 | 0.9790 | 0.9995 | 4.80 |
| | $\star$ DiffuSeq | 0.2413 | 0.5880 | 0.8365 | 0.9807 | 0.2732 | 2.60 |
| | $\star$ SeqDiffuSeq | 0.2434 | - | 0.8400 | 0.9807 | - | 2.33 |
| | $\dagger$ Dinoiser | 0.1949 | 0.5316 | 0.8036 | 0.9723 | 0.8643 | 6.20 |
| | Meta-Diffu$B$ | **0.2632** | **0.5933** | **0.8519** | **0.9902** | **0.2595** | **1.00** |
| WA | $\star$ GPT2-base | 0.3083 | 0.5461 | 0.8021 | 0.9439 | 0.5444 | 3.40 |
| | $\star$ GPT2-large | 0.2693 | 0.5111 | 0.7882 | 0.9464 | 0.6042 | 4.00 |
| | $\star$ LevT | 0.2052 | 0.4402 | 0.7254 | **0.9715** | 0.9907 | 5.00 |
| | $\star$ DiffuSeq | 0.3622 | 0.5849 | 0.8126 | 0.9264 | 0.4642 | 3.00 |
| | $\star$ SeqDiffuSeq | 0.3712 | - | 0.8214 | 0.9077 | - | 3.33 |
| | $\dagger$ Dinoiser | 0.2388 | 0.4821 | 0.6787 | 0.8421 | 0.9132 | 6.20 |
| | Meta-Diffu$B$ | **0.3877** | **0.6047** | **0.8233** | 0.9355 | **0.3888** | **1.60** |
| QT | $\star$ GPT2-base | 0.0741 | 0.2714 | 0.6052 | 0.9602 | **0.1403** | 3.80 |
| | $\star$ GPT2-large | 0.1110 | 0.3215 | **0.6346** | **0.9670** | 0.2910 | 2.60 |
| | $\star$ LevT | 0.0930 | 0.2893 | 0.5491 | 0.8914 | 0.9830 | 5.40 |
| | $\star$ DiffuSeq | 0.1731 | 0.3665 | 0.6123 | 0.9056 | 0.2789 | 3.20 |
| | $\star$ SeqDiffuSeq | 0.1746 | - | 0.6174 | 0.9248 | - | 3.33 |
| | $\dagger$ Dinoiser | 0.0477 | 0.1872 | 0.4690 | 0.8191 | 0.5273 | 6.40 |
| | Meta-Diffu$B$ | **0.1820** | **0.3870** | 0.6286 | 0.9323 | 0.2527 | **1.80** |
| CC | $\star$ GPT2-base | 0.0108 | 0.1508 | 0.5279 | 0.9194 | 0.0182 | 4.00 |
| | $\star$ GPT2-large | 0.0125 | 0.1002 | 0.5293 | 0.9244 | 0.0213 | 4.00 |
| | $\star$ LevT | 0.0158 | 0.0550 | 0.4760 | **0.9726** | 0.7103 | 3.80 |
| | $\star$ DiffuSeq | 0.0139 | 0.1056 | 0.5131 | 0.9467 | 0.0144 | 3.40 |
| | $\star$ SeqDiffuSeq | 0.0112 | - | 0.4425 | 0.9608 | - | 2.80 |
| | $\dagger$ Dinoiser | 0.0096 | 0.1166 | 0.3545 | 0.2485 | 0.9994 | 6.00 |
| | Meta-Diffu$B$ | **0.0220** | **0.1528** | **0.5316** | 0.9670 | **0.0133** | **1.20** |

# 7 Related Works

## 7.1 Text Diffusion

[11, 1] define an absorbing state for generating discrete data. Diffusion-LM [21] and AnalogBits [3] propose imposing noise on continuous latent representations, using transformation functions to bridge the discrete and continuous spaces of texts for both unconditional and controlled text generation.

## 7.2 Meta-Exploration

To transcend the limitations imposed by human-crafted rules in noise scheduling, we developed an additional model trained through Meta-Exploration, as inspired by [43]. Meta-Exploration is a Reinforcement Learning (RL) training method that utilizes learning effectiveness to devise sampling strategies that enhance model performance. Numerous studies [5, 37, 25, 19, 16] have employed Meta-Exploration to meta-learn scheduling strategies for applying additive Gaussian noise on actions and for sampling effective training data in RL tasks. We have adopted the Meta-Exploration concept [43] to train an additional model specifically for noise scheduling in S2S-Diffusion.

# 8 Broader Impact

In this work, our Meta-Diffu$B$ demonstrates significant performance improvements over previous S2S-Diffusion models across four Seq2Seq tasks, as detailed in Section 4.1. Meta-Diffu$B$ implements

Table 4: The results of our Meta-Diffu$B$ ($D_\theta$ = DiffuSeq) and other S2S-Diffusion models for generating sentences (E) and (H) on the WA dataset. The best result in each group is highlighted in **bold**.

| Methods | BLEU (↑) | Self-BLEU (↓) |
|---|---|---|
| DiffuSeq (E) | 0.3721 | 0.4345 |
| SeqDiffSeq (E) | 0.3752 | 0.4652 |
| Dinoiser (E) | 0.2892 | 0.8852 |
| Meta-Diffu$B$ (E) | **0.3997** | **0.3688** |
| DiffuSeq (H) | 0.3216 | 0.5085 |
| SeqDiffSeq (H) | 0.3282 | 0.6251 |
| Dinoiser (H) | 0.2092 | 0.9528 |
| Meta-Diffu$B$ (H) | **0.3724** | **0.4056** |

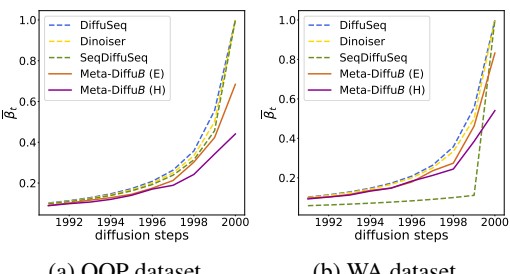

(a) QQP dataset      (b) WA dataset

Figure 3: Visualization of noise scheduling for each S2S-Diffusion model on the QQP and WA datasets. $\overline{\beta}_t$ represents the average noise imposed on sentences at diffusion step $t$. Unlike other models, which impose the same noise on all sentences, our Meta-Diffu$B$ ($D_\theta$ = DiffuSeq) varies the noise levels.

Table 5: Results of the plug-and-play experiment for our scheduler model. The 'Scheduler' field indicates the dataset used to train our scheduler model, while the 'DiffuSeq' field indicates the dataset used to train DiffuSeq. If the 'DiffuSeq' field is 'Null', DiffuSeq generates sentences using its own noise. Results that outperform those where DiffuSeq uses its own noise scheduling are highlighted in **bold**.

| Scheduler | DiffuSeq | BLEU (↑) | ROUGH-L (↑) | BERTScore (↑) | Dist-1 (↑) | Self-BLEU (↓) |
|---|---|---|---|---|---|---|
| WA |  | **0.2594** | **0.5912** | **0.8459** | **0.9834** | **0.2653** |
| QT | QQP | **0.2603** | **0.5947** | **0.8503** | **0.9812** | **0.2649** |
| Null |  | 0.2413 | 0.5880 | 0.8365 | 0.9807 | 0.2732 |

learnable, contextualized noise scheduling for Seq2Seq tasks. It not only shows enhanced generation quality and diversity but also has the potential to be applied to other diffusion models that require conditional data learning to generate target data. However, it is important to note that using Meta-Diffu$B$ to create fake news or other forms of misinformation is strongly discouraged.

# 9 Conclusions

We propose integrating Meta-Exploration into S2S-Diffusion models through our newly developed Meta-Diffu$B$. By utilizing Meta-Exploration to schedule contextualized noise, our Meta-Diffu$B$ model demonstrates significant performance improvements on four Seq2Seq benchmark datasets compared to previous S2S-Diffusion models and PLMs. We have conducted a comprehensive investigation of the noise-scheduling capabilities of Meta-Diffu$B$ and have visualized the results. Importantly, Meta-Diffu$B$ has the potential to act as a plug-and-play model, providing a promising approach for enhancing other S2S-Diffusion models during the inference stage without the need for fine-tuning.

# 10 Acknowledgments

We would like to express our sincere gratitude to Professor Hung-yi Lee from NTU Speech Lab for his invaluable guidance and insightful advice throughout this work. We are also deeply grateful to Professor Ray-I Chang from NTU ICAN Lab for his mentorship and constructive feedback. Additionally, we would like to thank the reviewers for their positive evaluation and valuable suggestions. Finally, we extend our appreciation to Maxora AI for providing the computational resources and environment that made this research possible, enabling us to make meaningful contributions to the field.

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

# A  Details of Datasets

There are four datasets in our experiment. For a fair comparison, we use the same datasets with the same settings as those described in [8, 45]. Below, we detail the datasets used:

- **Commonsense Conversation Dataset (CC) [48]:** CC requires models to generate informative responses given a dialogue context, an open-domain dialogue generation task. Extracted from Reddit single-round dialogues, it includes over 3 million conversational pairs. The training set contains 3,382,137 pairs, the development set has 2,048, and the test set includes 10,000 pairs. We train all S2S-Diffusions for 140,000 epochs on this dataset.
- **Quasar-T Dataset (QT) [6]:** QT requires models to generate questions given a context, a question-generation task. Extracted from Quasar-T, it consists of 119K training samples, with 116,953 in the training set, 2,048 in the development set, and 10,000 in the test set. We train all S2S-Diffusions for 40,000 epochs on this dataset.
- **Wiki-Auto Dataset (WA) [17]:** Wiki-Auto requires models to revise complex text into sequences with simplified grammar and vocabulary, a text simplification task. Extracted from Wikipedia, it includes 677K complex-simple sentence pairs with revision alignment. The training set contains 677,751 pairs, the development set has 2,048, and the test set has 80,000.
- **Quora Question Pairs Dataset (QQP) [8]:** QQP requires models to generate an alternative phrasing in the same language that conveys the same semantic content, a paraphrase generation task. Extracted from the Quora forum, it features 147K question-answering pairs. The training set contains 144,715 pairs, the development set has 2,048, and the test set has 2,500. We train all S2S-Diffusions for 140,000 epochs on this dataset.

# B  Detailed Information on Baselines

We provide details on the PLMs [33, 10] proposed by [8] and other S2S-Diffusion models [44, 45]. DiffuSeq fine-tunes PLMs to achieve optimal performance, balancing the trade-off between quality and diversity on the development set.

- **Dinoiser [44]:** In the training stage, Dinoiser proposes Clipping Threshold, a pre-defined rule to calculate the minimum value and then uses the $sqrt$ function to determine $\{\beta_1, ..., \beta_T\}$ for diffusing a batch of sentences in a training epoch. In the inference stage, Dinoiser deploys the noise from its latest training epoch to generate sentences.
- **SeqDiffuSeq [45]:** In the training stage, SeqDiffuSeq proposes imposing an increasing amount of noise with increasing training epochs. SeqDiffuSeq follows the pre-defined formula indicated in its paper to determine $\{\beta_1, ..., \beta_T\}$ for diffusing a batch of sentences in each training epoch. In the inference stage, SeqDiffuSeq deploys the noise from its latest training epoch to generate sentences.
- **Fine-Tuned GPT-2 Models [33]:** GPT2-base and GPT2-large are fine-tuned large pre-trained language models (PLMs) GPT2. GPT2-base's model parameter is 117M. GPT2-large's model parameter is 774M. DiffuSeq makes these two models inference with Beam Search and tunes the temperature to achieve better diversity.
- **LevT [10]:** LevT, a widely used, strong iterative NAR model. DiffuSeq sets the max iteration to 9 and follows the termination condition mentioned in the original paper. LevT's model parameterer is 80M.

# C  Network Architecture of Meta-Diffu$B$

Our Meta-Diffu$B$ framework comprises a scheduler model and an exploiter model. The exploiter model is an S2S-Diffusion model that adopts DiffuSeq's architecture, featuring a 12-layer Transformer with 12 attention heads. It incorporates time step embedding similarly to position embedding. The maximum sequence length is 128, and it operates over 2,000 diffusion steps. The scheduler model is an autoregressive model with a one-layer long short-term memory (LSTM) encoder-decoder architecture. It has the same embedding size as our exploiter model, with a maximum encoder length of 128 and a decoder length of 2,000.

# D  Details of Evaluation Metrics

We demonstrate the details of the evaluation metrics used in this work. To ensure a robust comparison, we adhere to the same evaluation metric settings as previous S2S-Diffusion models [8, 45].

- **BLEU**[27]: BLEU is widely used to measure the quality of text generation. In this work, we use a smoothed BLEU score ranging from BLEU-1 to BLEU-4, where higher scores indicate better quality.

- **ROUGE-L** [2]: ROUGE-L measures text generation quality by calculating the longest common subsequence. Higher ROUGE-L scores indicate better quality.

- **BERTScore** [46]: BERTScore assesses text generation quality by calculating the embedding similarity between generated sentences and reference sentences. It utilizes embeddings from a BERT model, with higher scores indicating better quality.

- **Dist-1** [2]: Dist-1 measures the diversity of generated text by calculating the uniqueness of words within a single generated target sentence.

- **Self-BLEU** [27]: Self-BLEU assesses the diversity of text generation by calculating the ratio of unique 4-grams in a set of generated sentences. Lower Self-BLEU scores indicate better diversity.

# E  Experiments of Meta-Diffu$B$ on Machine Translation and Other Datasets

Following Section 5, we conduct experiments on machine translation datasets, including IWSLT14 DE-EN [26] and WMT14 DE-EN [31], using the same dataset and evaluation metric settings as other S2S-Diffusion models [45, 44]. Specifically, we adopt SacreBLEU [32] as the evaluation metric. As shown in Table 7, our Meta-Diffu$B$ improves the performance of DiffuSeq, Dinoiser, and SeqDiffuSeq on these machine translation tasks. Additionally, we provide experiments on DiffuSeq-based S2S-Diffusion models and discrete S2S-Diffusion models (RDM [47] based on D3PM [1]) on the QQP and QG datasets. As shown in Table 7, our Meta-Diffu$B$ consistently improves performance across both discrete and DiffuSeq-based S2S-Diffusion models.

Table 6: Results of Meta-Diffu$B$ on Machine Translation datasets (DE-EN). Results where Meta-Diffu$B$ combined with different models show improved performance are indicated in **bold**.

| Methods | SacreBLEU ↑ (IWSLT14 DE-EN) | SacreBLEU ↑ (WMT14 DE-EN) |
|---|---|---|
| DiffuSeq | 29.43 | 22.72 |
| Meta-Diffu$B$ ($D_\theta$ = DiffuSeq) | **31.71** | **26.17** |
| SeqDiffuSeq | 30.16 | 23.28 |
| Meta-Diffu$B$ ($D_\theta$ = SeqDiffuSeq) | **32.41** | **26.14** |
| Dinoiser | 31.61 | 30.30 |
| Meta-Diffu$B$ ($D_\theta$ = Dinoiser) | **33.82** | **32.09** |

Table 7: Comparison of Meta-Diffu$B$ on the QG and QQP datasets. Results where Meta-Diffu$B$ combined with different models show improved performance are indicated in **bold**.

| Methods | BLEU ↑ (QQP) | BERTScore ↑ (QQP) | BLEU ↑ (QG) | BERTScore ↑ (QG) |
|---|---|---|---|---|
| DiffuSeq [8] | 0.2413 | 0.8365 | 0.1731 | 0.6123 |
| Meta-Diffu$B$ ($D_\theta$ = DiffuSeq) | **0.2552** | **0.8821** | **0.1826** | **0.6357** |
| DiffuSeq-v2 [9] | 0.2411 | 0.8393 | - | - |
| Meta-Diffu$B$ ($D_\theta$ = DiffuSeq-v2) | **0.2556** | **0.8829** | - | - |
| BG-DiffuSeq [42] | 0.2619 | 0.8427 | 0.1744 | 0.6280 |
| Meta-Diffu$B$ ($D_\theta$ = BG-DiffuSeq) | **0.2790** | **0.8757** | **0.1838** | **0.6571** |
| TESS [22] | 0.3020 | 0.8570 | 0.1950 | 0.6580 |
| Meta-Diffu$B$ ($D_\theta$ = TESS) | **0.3142** | **0.8975** | **0.2055** | **0.6761** |
| RDM [47] | 0.2510 | 0.8472 | 0.1802 | 0.6310 |
| Meta-Diffu$B$ ($D_\theta$ = RDM) | **0.2684** | **0.8724** | **0.2271** | **0.6542** |

# F Showcase of Generated Sentences

Our Meta-Diffu$B$ ($D_\theta$ = DiffuSeq) achieves better generation diversity and quality on the hardest generated sentences (H), as evidenced by the examples provided in Table 8 and Table 9. These tables illustrate that Meta-Diffu$B$ can generate more effective sentences (H) from both the WA and QQP datasets than other S2S-Diffusion models. Table 8 shows the performance of our Meta-Diffu$B$ ($D_\theta$

Table 8: The sample output of our Meta-Diffu$B$ ($D_\theta$ = DiffuSeq) and other S2S-Diffusion models [8, 44, 45] on hardest generated sentences (H) of WA dataset. The conditional sentence is the same.

| **Conditional sentence**: ostersunds bs is a bandy club in ostersund, sweden, established on 5 september 1974 when ope ifs bandy section was disestablished. | | | | |
|---|---|---|---|---|
| **Real target sentence** | **Meta-Diffu$B$** | **DiffuSeq** | **Dinoiser** | **SeqDiffuSeq** |
| it was established on 5 september 1974 when ope ifs bandy section was disestablished. | it was established on 5 september 1974 when ope if ifs bandy section was disestablished. | ostersunds bs is a bandy club in the town of ostersund in sweden. | The club was disestablished on 5 September 1974. | henry cr bandy club in the town of sweden. |
| | ostersunds bs is a bandy club started in ostersund, sweden, established in 5 september 1974, the ifs bandy section was disestablished. | ostersunds bs is a bandy club in the town of ostersund in sweden. | The club was disestablished on 5 September 1974. | henry cr bandy club in the town of sweden. |
| | it was founded on 5 september 1974 when ope ifs bandy section was disestablished. | ostersunds bs is a bandy club in the town of ostersund in sweden. | The club was disestablished on 5 September 1974. | henry cr bandy club in the town of sweden. |

= DiffuSeq). It consistently generates the hardest sentences (H) with superior quality and diversity, providing a reliable solution. In contrast, other S2S-Diffusion models [8, 44, 45] often produce repetitive sentences, failing to ensure both quality and diversity. Our findings, as illustrated in Table 9, show that Meta-Diffu$B$ ($D_\theta$ = DiffuSeq) also outperforms other models, generating sentences (H) with superior diversity and quality on the QQP dataset.

Table 9: The sample output of Meta-Diffu$B$ ($D_\theta$ = DiffuSeq) and other S2S-Diffusion models [8, 44, 45] on hardest generated sentence (H) of QQP dataset. The conditional sentence is the same.

| **Conditional sentence**: is it possible to invent the time machine? | | | | |
|---|---|---|---|---|
| **Real target sentence** | **Meta-Diffu$B$** | **DiffuSeq** | **Dinoiser** | **SeqDiffuSeq** |
| can we create a time machine? | is we really possible to create a time machine? | is it possible to in our time machine? | Is it possible to make a time machine? | is possible to have a time machine? |
| | is it possible that we create a time machine? | is it possible to in our time machine? | Is it possible to make a time machine? | is possible to have a time machine? |
| | can we create a time machine? | is it possible to in our time machine? | Is it possible to make a time machine? | is possible to have a time machine? |

# G More Metrics on Contextualized Noise Scheduling of Meta-Diffu$B$

We present the results of Table 4 evaluated by other metrics in Table 10. Table 10 illustrates that our Meta-Diffu$B$ ($D_\theta$ = DiffuSeq) outperforms other S2S-Diffusion models in generating sentences (H) and (E) across all evaluation metrics in this study.

# H Context-Aware Noise Generation in Meta-Diffu$B$: Analysis and Insights

In this section, we further discuss the noise generated by our Meta-Diffu$B$. As shown in Figure 4, the total noise per epoch produced by Meta-Diffu$B$ exhibits less fluctuation compared to other rule-based methods. While the noise variance is smaller than that of SeqDiffuSeq, it is larger than Dinoiser. From Table 3, we can see that the magnitude or variability of the noise does not necessarily correlate with

Table 10: The results of our Meta-Diffu$B$ and other S2S-Diffusion models [8, 44, 45] for generating sentences (E) and (H) on the WA dataset. The best result in each group is highlighted in **bold**.

| Methods | BLEU (↑) | Self-BLEU (↓) | ROUGH-L (↑) | BERTScore (↑) | Dist-1 (↑) |
|---|---|---|---|---|---|
| DiffuSeq (E) | 0.3721 | 0.4345 | 0.5962 | 0.8232 | 0.9285 |
| SeqDiffSeq (E) | 0.3752 | 0.4652 | 0.6021 | 0.8286 | 0.9273 |
| Dinoiser (E) | 0.2892 | 0.8852 | 0.4937 | 0.6882 | 0.8574 |
| Meta-Diffu$B$ (E) | **0.3997** | **0.3688** | **0.6359** | **0.8452** | **0.9462** |
| DiffuSeq (H) | 0.3216 | 0.5085 | 0.5514 | 0.7586 | 0.8828 |
| SeqDiffSeq (H) | 0.3282 | 0.6251 | 0.5621 | 0.7479 | 0.8974 |
| Dinoiser (H) | 0.2092 | 0.9528 | 0.4375 | 0.6345 | 0.8396 |
| Meta-Diffu$B$ (H) | **0.3724** | **0.4056** | **0.5741** | **0.8026** | **0.9216** |

better performance for S2S-Diffusion models. The noise generated by Meta-Diffu$B$ is context-aware, meaning it is learned and tailored to each sentence, thereby achieving better performance.

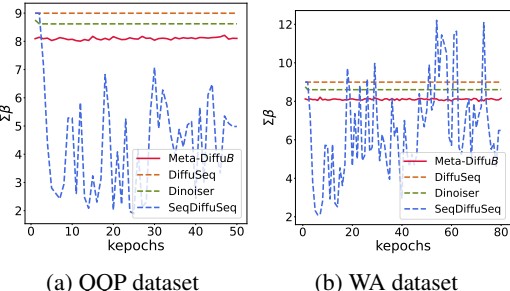

(a) QQP dataset          (b) WA dataset

Figure 4: Adaptive noise scheduling for each S2S-Diffusion model on the QQP and WA datasets. $\Sigma\beta$ represents the total amount of noise imposed in each training epoch.

# I   Additional Plug-and-Play Experiments with the Scheduler Model

We present the results of the plug-and-play experiment featuring our scheduler model and DiffuSeq [8] trained on different datasets in Table 11. This table demonstrates that our scheduler model can enhance the performance of DiffuSeq across four datasets during the inference stage without the need for fine-tuning.

Table 11: Results of the plug-and-play experiment for our scheduler model. The 'Scheduler' field indicates the dataset used to train the scheduler model, while the 'DiffuSeq' field indicates the dataset used to train DiffuSeq. If the 'DiffuSeq' field is 'Null', DiffuSeq generates sentences using its own noise. Results that outperform those where DiffuSeq uses its own noise scheduling are highlighted in **bold**.

| Scheduler | DiffuSeq | BLEU (↑) | ROUGH-L (↑) | BERTScore (↑) | Dist-1 (↑) | Self-BLEU (↓) |
|---|---|---|---|---|---|---|
| WA | | **0.2594** | **0.5912** | **0.8459** | **0.9834** | **0.2653** |
| QT | QQP | **0.2603** | **0.5905** | **0.8503** | **0.9812** | **0.2649** |
| Null | | 0.2413 | 0.5880 | 0.8365 | 0.9807 | 0.2732 |
| WA | | **0.1804** | **0.3761** | **0.6234** | **0.9147** | 0.3848 |
| QQP | QT | **0.1769** | **0.3729** | **0.6215** | **0.9104** | 0.4485 |
| Null | | 0.1731 | 0.3665 | 0.6123 | 0.9056 | **0.2789** |
| QT | | **0.3666** | **0.5945** | **0.8217** | **0.9297** | **0.4052** |
| QQP | WA | **0.3711** | **0.5985** | **0.8204** | **0.9302** | **0.3996** |
| Null | | 0.3622 | 0.5849 | 0.8126 | 0.9264 | 0.4642 |

We also provide experiments with our scheduler on various other S2S-Diffusion models. As shown in Table 12 and Table 13, our scheduler not only improves performance across datasets but also enhances the performance across different models.

Table 12: Plug-and-play experiments on SeqDiffuSeq integrated with our scheduler. The field 'SeqDiffuSeq' indicates which dataset this model is trained on. When the 'Scheduler' field is 'Null', it indicates the use of the model's own noise scheduling. Results where the model performs better with its own noise are indicated in **bold**.

| Scheduler | SeqDiffuSeq | BLEU ↑ | BERTScore ↑ | Dist-1 ↑ |
|---|---|---|---|---|
| WA | QQP | 0.2627 | 0.8481 | 0.9814 |
| Null | QQP | 0.2434 | 0.8400 | 0.9807 |
| WA | QT | 0.1834 | 0.6226 | 0.9369 |
| Null | QT | 0.1746 | 0.6174 | 0.9248 |

Table 13: Plug-and-play experiments on Dinoiser integrated with our scheduler. The field 'Dinoiser' indicates which dataset this model is trained on. When the 'Scheduler' field is 'Null', it indicates the use of the model's own noise scheduling. Results where the model performs better with its own noise are indicated in **bold**.

| Scheduler | Dinoiser | BLEU ↑ | BERTScore ↑ | Dist-1 ↑ |
|---|---|---|---|---|
| WA | QQP | 0.2079 | 0.8121 | 0.9765 |
| Null | QQP | 0.1949 | 0.8036 | 0.9723 |
| WA | QT | 0.0495 | 0.4740 | 0.8289 |
| Null | QT | 0.0477 | 0.4690 | 0.8191 |

