# OpenReview forum: "Meta-Diffu$B$: A Contextualized Sequence-to-Sequence Text Diffusion Model with Meta-Exploration"
_NeurIPS.cc/2024/Conference — NeurIPS 2024 poster_

### Official Review · Reviewer_qgkV · 2024-06-28

**Soundness:** 2
**Presentation:** 2
**Contribution:** 2
**Rating:** 4
**Confidence:** 3

**Summary:**

This paper presents a novel approach to improving sequence-to-sequence (Seq2Seq) text generation models using diffusion models. The authors identify limitations in existing Seq2Seq-Diffusion models, which typically rely on fixed or hand-crafted noise scheduling rules that do not account for the specific characteristics of Seq2Seq tasks. To address these limitations, they propose the Meta-DiffuB framework, which introduces a scheduler-exploiter paradigm designed to provide contextualized noise scheduling. The scheduler model dynamically schedules contextualized noise based on the semantics of each sentence. The scheduler model is trained using Meta-Exploration techniques to optimize noise scheduling and can function as a "plug-and-play" model. The exploiter model (an S2S-Diffusion model) utilizes the noise scheduled by the scheduler for updating and text generation.

**Strengths:**

- The introduction of a scheduler-exploiter framework to schedule contextualized noise for each sentence in Seq2Seq tasks is a departure from existing methods that use fixed or hand-crafted noise scheduling rules. This idea is intuitive and straightforward.
- Experimentation on four benchmark Seq2Seq datasets somehow validates the effectiveness of the proposed Meta-DiffuB framework.
- The paper follows a logical structure, starting with the introduction and problem statement, followed by the methodology, experiments, results, and conclusions.

**Weaknesses:**

# More Comprehensive Comparison
My first concern is that this paper mainly compares the proposed method with Diffuseq, SeqDiffuSeq, and Dinoiser. There are many other diffusion models tailored for general seq2seq tasks [1-6]. I would suggest the authors carefully review the existing literature and select more recent baselines to compare.

1. Empowering Diffusion Models on the Embedding Space for Text Generation. *NAACL*
2. Latent Diffusion for Language Generation. *NeurIPS*
3. DiffuSeq-v2: Bridging Discrete and Continuous Text Spaces for Accelerated Seq2Seq Diffusion Models. *EMNLP*
4. AR-Diffusion: Auto-Regressive Diffusion Model for Text Generation. *NeurIPS*
5. Can Diffusion Model Achieve Better Performance in Text Generation? Bridging the Gap between Training and Inference! *EMNLP*
6. TESS: Text-to-Text Self-Conditioned Simplex Diffusion. *EACL*

The authors may also find more related works in:
- https://github.com/StevenYuan666/Awesome-Diffusion-Models-for-NLP
- https://github.com/westfish/Awesome-NLP-Diffusion-Models
- https://github.com/AoiDragon/Awesome-Text-Diffusion-Models

Also, some more recent works focus on the noise schedule:

7. Effective Integration of Text Diffusion and Pre-Trained Language Models with Linguistic Easy-First Schedule. *LREC-COLING*

# Presentation
- As the paper mentioned, parts of the methodology are motivated by the Meta-Exploration. I would suggest the authors add a section in the preliminary to provide more necessary details about the meta-exploration techniques, which will benefit readers who are not familiar with Meta-Exploration.

# Analysis
- The authors claim that the proposed method results in a contextualized noise schedule, which makes sense to me in terms of the scheduler generating Meta-Instructions conditioned on $\boldsymbol{w}^x$. However, from Figure 3, even though the total amount of noise indeed fluctuates across different training epochs, it always keeps at the same level and the differences seem insignificant. I'm wondering why this makes the proposed methods beat existing baselines.

**Questions:**

1. Following my argument above, can you provide a deeper interpretation of the noise scheduling visualizations and their implications for different types (easy or hard as mentioned in the paper) of sentences? How do different noise schedules affect the generation quality and diversity for various types of sentences? Any patterns or insights observed from these visualizations?
2. Could you provide an intuitive explanation of why the scheduler can function as a plug-and-play model? From the analysis in section 4.7, I'm suspecting it is because the learned schedule is not quite different from the pre-defined noise schedule.
3. Did you run the empirical evaluation for several runs by setting different random seeds? If so, could you also highlight which results are statistically significant than the baseline method?

**Limitations:**

The proposed method is in the domain of text generation and does not directly have any negative impact by itself. The authors have provided a paragraph to discuss the negative potential of misinformation created by the text generation model.

---

> ### Author Response · Authors · 2024-08-07
> **Thank you for your valuable suggestions. We have included additional baseline model experiments and addressed each of your questions to clarify any doubts. We assure you that these experiments will be included in the final version.**
>
> >Q1: My first concern is that this paper mainly compares the proposed method with Diffuseq, SeqDiffuSeq, and Dinoiser. There are many other diffusion models tailored for general seq2seq tasks [1-6]. I would suggest the authors carefully review the existing literature and select more recent baselines to compare.
>
> **Response:**: Thank you for the insightful feedback. We have included additional baselines in the following Table, specifically those that are evaluated on the same dataset as ours and have open-source implementations (Diffuseq-v2, BG-DiffuSeq and TESS). We demonstrate that our Meta-Diffu$B$ framework can integrate various baselines as exploiter models, achieving superior performance compared to the original models. This is consistent with the experiments and results presented in Appendix F.
>
> **More recent baselines compared with our Meta-Diffu$B$ on QG and QQP datasets**
> |Methods|BLEU ↑ (QQP)|BERTScore ↑ (QQP)|BLEU↑ (QG)|BERTScore ↑ (QG)|
> |-|-|-|-|-|
> |DiffuSeq|0.2413|0.8365|0.1731|0.6123|
> |Meta-Diffu$B$ (exploiter=DiffuSeq)|**0.2552**|**0.8821**|**0.1826**|**0.6357**|
> |DiffuSeq-v2 |0.2411|0.8393|-|-|
> |Meta-Diffu$B$ (exploiter=DiffuSeq-v2)|**0.2556**|**0.8829**|-|-|
> |BG-DiffuSeq |0.2619|0.8427|0.1744|0.6280|
> |Meta-Diffu$B$ (exploiter=BG-DiffuSeq)|**0.2790**|**0.8757**|**0.1838**|**0.6571**|
> |TESS |0.3020|0.8570|0.1950|0.6580|
> |Meta-Diffu$B$ (exploiter=TESS)|**0.3142**|**0.8975**|**0.2055**|**0.6761**|
>
> >Q2: As the paper mentioned, parts of the methodology are motivated by Meta-Exploration. I would suggest the authors add a section in the preliminary to provide more necessary details about the meta-exploration techniques, which will benefit readers who are not familiar with Meta-Exploration.
>
> **Response:** To assist readers in understanding the foundational concepts, we will move the discussion of Meta-Exploration from the Related Work section to the Preliminary section in our final version.
>
> >Q3: The authors claim that the proposed method results in a contextualized noise schedule, which makes sense to me in terms of the scheduler generating Meta-Instructions conditioned on. However, from Figure 3, even though the total amount of noise indeed fluctuates across different training epochs, it always keeps at the same level and the differences seem insignificant. I'm wondering why this makes the proposed methods beat existing baselines.
>
> **Response:**
> In Figure 3, our method differs from approaches like SeqDiffuSeq or Dinoiser, which rely on predefined rules. Instead, our approach is data-driven, with the model learning the underlying patterns. As a result, the degree of variation—whether large or small—is not the primary focus. Additionally, the learned noise can be flexibly integrated into other systems in a plug-and-play manner.
>
> >Q4: Can you provide a deeper interpretation of the noise scheduling visualizations and their implications for different types (easy or hard as mentioned in the paper) of sentences? How do different noise schedules affect the generation quality and diversity for various types of sentences? Any patterns or insights observed from these visualizations?
>
> **Response:** From the visualizations, we observe that the contextualized noise scheduling mechanism helps in maintaining a delicate balance between exploration (diversity) and exploitation (quality). The patterns indicate that our model has learned different strategies for different types of sentences. These insights demonstrate the effectiveness of our Meta-Diffu$B$ framework in handling various sentence complexities, resulting in superior performance across multiple datasets.
>
> >Q5: Could you provide an intuitive explanation of why the scheduler can function as a plug-and-play model? From the analysis in section 4.7, I'm suspecting it is because the learned schedule is not quite different from the pre-defined noise schedule.
>
> Our scheduler model learns directly from sentence semantics, allowing it to provide learnable noise across different text datasets. Figure 3 shows that Dinoiser and SeqDiffuSeq rely on rule-based noise variations, which do not necessarily lead to better performance.
>
>
> >Q6: Did you run the empirical evaluation for several runs by setting different random seeds? If so, could you also highlight which results are statistically significant than the baseline method?
>
> **Response:**  Thank you for your question regarding the empirical evaluation with different random seeds. We followed the evaluation protocols of DiffuSeq and Diffusion-LM, running multiple experiments with different random seeds. In Tables 1, 2, and 3, we used MBR, so the results in bold are all statistically significant compared to the baseline methods.

---

> > ### Comment · Reviewer_qgkV · 2024-08-10
> > **Thank you**
> >
> > Thank you for your detailed response. I have read through most of the comments by other reviewers, as well as the rebuttal. I decide to keep my original score.

---

> > > ### Author Response · Authors · 2024-08-14
> > > **Thank you very much for your suggestion. We have also supplemented the experiment you mentioned.**
> > >
> > > In addition to the baselines we previously supplemented, we have also included the easy-first schedule and the discrete S2S-Diffusion model you mentioned. The discrete S2S-Diffusion models, which rely on absorbing states, still require a set of $\beta$ values. Therefore, our Meta-Diffu$B$ can still be directly applied to the discrete S2S-Diffusion model. The paper on the easy-first schedule, Diffusion-LEF, did not implement the method on the same dataset as ours. Hence, we applied this noise schedule to DiffuSeq without using a pre-trained BERT model to ensure fairness. Below is the table summarizing the additional experiments we conducted.
> > >
> > > **We compared more recent baselines with our Meta-Diffu$B$ on the QG and QQP datasets. These baselines include both discrete and continuous S2S-Diffusion models.**
> > > |Methods|BLEU ↑ (QQP)|BERTScore ↑ (QQP)|BLEU↑ (QG)| BERTScore ↑ (QG)|
> > > |-|-|-|-|-|
> > > |DiffuSeq|0.2413|0.8365|0.1731|0.6123|
> > > |Diffusion (Easy-First Schedule)|0.2503|0.8692|0.1812|0.6253|
> > > |Meta-Diffu$B$ (exploiter=DiffuSeq)|**0.2552**| **0.8821**|**0.1826**|**0.6357**|
> > > |DiffuSeq-v2|0.2411|0.8393|-|-|
> > > |Meta-Diffu$B$ (exploiter=DiffuSeq-v2)|**0.2556**| **0.8829**|-|-|
> > > |BG-DiffuSeq|0.2619|0.8427|0.1744|0.6280|
> > > |Meta-Diffu$B$ (exploiter=BG-DiffuSeq)|**0.2790**| **0.8757**|**0.1838**|**0.6571**|
> > > |TESS|0.3020|0.8570|0.1950|0.6580|
> > > |Meta-Diffu$B$ (exploiter=TESS)|**0.3142**|**0.8975**| **0.2055**|**0.6761**|
> > > |RDM (Discrete Diffusion)|0.2510|0.8472|0.1802|0.6310|
> > > |Meta-Diffu$B$ (exploiter=RDM)|**0.2684**|**0.8724**|**0.2271**|**0.6542**|
> > >
> > > As shown in the Table, when combined with the discrete S2S-Diffusion model RDM, we also achieve satisfactory results. Additionally, the noise generated by the scheduler through learning, compared to the noise produced by the easy-first schedule that relies on many rules, can further enhance the performance of DiffuSeq.

---

### Official Review · Reviewer_5aL8 · 2024-07-12

**Soundness:** 3
**Presentation:** 3
**Contribution:** 2
**Rating:** 4
**Confidence:** 3

**Summary:**

Comprehensive Evaluation of the new S2S Diffusion framework Meta-Diffu$\beta$.

**Strengths:**

- Clear and well-written presentation of the method
- It provides a new framework that uses an additional scheduler based on Meta-Exploration to schedule contextualized noise, which performs well in the four given benchmarks.

**Weaknesses:**

- **Incremental Innovation**: In terms of the scheduler, the use of the Meta-Exploration method for scheduling contextualized noise is noted. However, I concern that the exploiters fully follow the DiffuSeq work, resulting in a lack of significant innovation.
- **Missing Experimental Baselines**: The cited baselines are all from before February 2023. It would be beneficial to include more recent baseline results [1] [2] [3] and take both generation efficiency and training speed into evaluation metrics.
- **Dataset Limitations**: The selected dataset completely follows DiffuSeq and does not take into account machine translation tasks and related datasets from SeqDiffuSeq and DINOISER. It would be advantageous to include test results on IWSLT14 and WMT14.
- **Plug-and-Play Capability**: The study only demonstrates the plug-and-play model and its corresponding effects on DiffuSeq. It's recommended to show the plug-and-play capability in other S2S Diffusion models like SeqDiffuSeq, DINOISER and other baselines mentioned to prove the claim in the conclusion.

Minor:

- The main text introduces the datasets but does not provide information about the downstream Seq2Seq tasks. Consider moving the introduction of these tasks from the appendix to the main text to enhance readability.

- The pre-trained model and environment dependencies are missing from your GitHub repository. It would be helpful to include these components to facilitate reproducibility.

[1] Ding, Y., Tian, J., Mei, S., Zhou, Y., Dong, Y., He, H. and Hu, W., 2023, December. LDSeq: Latent Diffusion Models for Sequence to Sequence Text Generation. In *Proceedings of the 2023 7th International Conference on Computer Science and Artificial Intelligence*

[2] Gong, S., Li, M., Feng, J., Wu, Z. and Kong, L., 2023. Diffuseq-v2: Bridging discrete and continuous text spaces for accelerated seq2seq diffusion models. *arxiv preprint arxiv:2310.05793*.

[3] Tang, Z., Wang, P., Zhou, K., Li, J., Cao, Z. and Zhang, M., 2023. Can Diffusion Model Achieve Better Performance in Text Generation? Bridging the Gap between Training and Inference!. *arxiv preprint arxiv:2305.04465*.

**Questions:**

- Regarding the selection of fixed noise, have you attempted to use different mathematical functions to obtain the $\beta$ values as inputs for the skipping function? How is the performance using different $\beta$ in the scheduler?

**Limitations:**

The authors have addressed the limitations.

---

> ### Author Response · Authors · 2024-08-07
> **Thank you for your valuable suggestions. We have included additional baseline model experiments, machine translation experiments, and a variety of different baselines. We have addressed each of your questions to clarify any doubts. We assure you that these experiments will be included in the final version.**
>
> >Q1: Incremental Innovation: In terms of the scheduler, the use of the Meta-Exploration method for scheduling contextualized noise is noted. However, I concern that the exploiters fully follow the DiffuSeq work, resulting in a lack of significant innovation.
>
> **Response**: Thank you for your feedback. As noted in Appendix F, our framework can utilize different diffusion models as exploiters, demonstrating its versatility. The key innovation is integrating these models with our scheduler, significantly enhancing performance. We propose a general method that strengthens other S2S-diffusion models, providing more than incremental improvements. In Sections 4.5 to 4.7, we compare our Meta-Diffu$B$, using DiffuSeq as the exploiter model, with other S2S-Diffusion models that incorporate adaptive noise to highlight its effectiveness.
>
> >Q2: Missing Experimental Baselines: The cited baselines are all from before February 2023. It would be beneficial to include more recent baseline results [1] [2] [3] and take both generation efficiency and training speed into evaluation metrics.
>
> **Response:** Thank you for the insightful feedback. We have included additional baselines in the following Table, specifically those that are evaluated on the same dataset as ours and have open-source implementations (Diffuseq-v2, BG-DiffuSeq and TESS). Results where Meta-Diffu$B$ combined with different models show improved performance are indicated in bold. Our Meta-Diffu$B$ framework can integrate various baselines as exploiter models, achieving superior performance compared to the original models. This is consistent with the experiments and results presented in Appendix F.
>
> **More recent baselines compared with our Meta-Diffu$B$ on QG and QQP datasets**
> |Methods|BLEU ↑ (QQP)|BERTScore ↑ (QQP)|BLEU↑ (QG)|BERTScore ↑ (QG)|
> |-|-|-|-|-|
> |DiffuSeq|0.2413|0.8365|0.1731|0.6123|
> |Meta-Diffu$B$ (exploiter=DiffuSeq)|**0.2552**|**0.8821**|**0.1826**|**0.6357**|
> |DiffuSeq-v2|0.2411|0.8393|-|-|
> |Meta-Diffu$B$ (exploiter=DiffuSeq-v2)|**0.2556**|**0.8829**|-|-|
> |BG-DiffuSeq|0.2619|0.8427|0.1744|0.6280|
> |Meta-Diffu$B$ (exploiter=BG-DiffuSeq)|**0.2790**|**0.8757**|**0.1838**|**0.6571**|
> |TESS|0.3020|0.8570|0.1950|0.6580|
> |Meta-Diffu$B$ (exploiter=TESS)|**0.3142**|**0.8975**|**0.2055**|**0.6761**|
>
> >Q3: Dataset Limitations: The selected dataset completely follows DiffuSeq and does not take into account machine translation tasks and related datasets from SeqDiffuSeq and DINOISER. It would be advantageous to include test results on IWSLT14 and WMT14.
>
> **Response:** Thank you for the valuable feedback. We supplement additional machine translation datasets in the following table and employed the same evaluation metrics used by SeqDiffuSeq and Dinoiser for these tasks. Results where Meta-Diffu$B$ combined with different models show improved performance are indicated in bold. Furthermore, we demonstrate that our Meta-Diffu$B$ framework, when combined with different S2S-Diffusion models, achieves superior performance in machine translation tasks as well.
>
> **Experiment of Meta-Diffu$B$ on Machine Translation dataset**
> |Methods|SacreBLEU ↑ (IWSLT14 DE-EN)|SacreBLEU ↑ (WMT14 DE-EN)|
> |-|-|-|
> |DiffuSeq|29.43|22.72|
> |Meta-Diffu$B$ (exploiter=DiffuSeq)|**31.71**|**26.17**|
> |SeqDiffuSeq|30.16|23.28|
> |Meta-Diffu$B$ (exploiter=SeqDiffuSeq)|**32.41**|**26.14**|
> |Dinoiser|31.61|30.30|
> |Meta-Diffu$B$ (exploiter=Dinoiser)|**33.82**|**32.09**|
>
> >Q4: Plug-and-Play Capability: The study only demonstrates the plug-and-play model and its corresponding effects on DiffuSeq. It's recommended to show the plug-and-play capability in other S2S Diffusion models like SeqDiffuSeq, DINOISER, and other baselines mentioned to prove the claim in the conclusion.
>
> **Response:** Thank you for your suggestion. We supplement our experiments with plug-and-play capabilities on Dinoiser and SeqDiffuSeq in the following tables. Our scheduler, trained on different datasets, improves the performance of both Dinoiser and SeqDiffuSeq. The fields "Dinoiser" and "SeqDiffuSeq" indicate which dataset these two models are trained on.  When the scheduler field is null, it indicates the use of the model's own noise scheduling. Results where the model performs better with its own noise are indicated in bold.
>
> **Plug-and-play experiments on SeqDiffuSeq integrated with our scheduler**
> |Scheduler|SeqDiffuSeq|BLEU ↑|BERTScore ↑|Dist-1 ↑ |
> |-|-|-|-|-|
> |WA|QQP|**0.2627**|**0.8481**|**0.9814**|
> |Null|QQP|0.2434|0.8400|0.9807|
> |WA|QT|**0.1834**|**0.6226**|**0.9369**|
> |Null|QT|0.1746|0.6174|0.9248|
>
> **Plug-and-play experiments on Dinoiser integrated with our scheduler**
> |Scheduler|Dinoiser|BLEU ↑|BERTScore ↑|Dist-1 ↑|
> |-|-|-|-|-|
> |WA|QQP|**0.2079**|**0.8121**|**0.9765**|
> |Null|QQP|0.1949|0.8036|0.9723|
> |WA|QT|**0.0495**|**0.4740**|**0.8289**|
> |Null|QT|0.0477|0.4690|0.8191|

---

### Official Review · Reviewer_YnwD · 2024-07-13

**Soundness:** 3
**Presentation:** 3
**Contribution:** 3
**Rating:** 6
**Confidence:** 4

**Summary:**

The paper introduces the Meta-DiffuB, a scheduler-exploiter diffusion framework that focuses on sequence-to-sequence (Seq2Seq) setting. Its novel trainable contextualized noise scheduler, inspired by Meta-exploration, is also flexible and plug-and-play with other models like DiffuSeq without re-training. This approach dynamically schedules noise levels based on the characteristics of each sentence, improving the performance of text generation tasks. Meta-DiffuB outperforms existing models and fine-tuned pre-trained language models across four Seq2Seq benchmark datasets, demonstrating the effectiveness of its adaptive noise scheduling.

**Strengths:**

- The paper is well-motivated and well-structured.
- The introduction of a contextualized noise scheduling strategy using Meta-Exploration is a significant advancement, addressing limitations of fixed or non-contextualized noise schedules in existing diffusion models.
- Meta-DiffuB shows superior performance across four Seq2Seq benchmark datasets.
- The ability to integrate the scheduler model into existing Seq2Seq diffusion models without the need for fine-tuning during inference enhances the practicality and usability of the approach.
- The paper provides details of the proposed framework, pseudo-code with actual code for ease of reproduction.

**Weaknesses:**

- While the model performs well on benchmark datasets, there is limited discussion on its scalability to larger, real-world datasets and more complex text generation tasks.
- The dynamic noise scheduling process might introduce additional computational overhead during inference, which is not thoroughly analyzed in the paper.
- Meta-DiffuB is built upon and compared with S2S-Diffusion models known as continuous-based text diffusion models. It lacks a detailed comparison with other discrete-based text diffusion models such as D3PM [1] or RDM [2]. RDM claimed to achieve better performance and inference speed compared to DiffuSeq.
- The effects of the dynamic noise scheduling on the overall training dynamics and convergence rate of the model are not explored in detail, which could be crucial for understanding the method’s effectiveness.

**Questions:**

- L120, 123: the quote for 'skipping' is wrongly formatted
- L202-203: it should be either "The maximum Minimum Bayes Risk (MBR)..." or "The Minimum Bayes Risk (MBR)....", also it cites paper [20] (the Rouge package) which I think is incorrect for MBR.
- Besides the increased training time compared to DiffuSeq (L217-218), it would be beneficial to know the increased inference time too.
- How well does the Meta-DiffuB model generalize to other types of text generation tasks beyond the Seq2Seq framework? Have experiments been conducted to evaluate its performance in different text generation scenarios?
- How does the model scale with larger datasets or more complex text generation tasks? Are there any specific optimizations or modifications required to handle real-world data volumes?

[1] https://arxiv.org/pdf/2107.03006

[2] https://arxiv.org/pdf/2302.05737

**Limitations:**

Yes

---

> ### Author Response · Authors · 2024-08-07
> **Thank you for your valuable suggestions. We have included the inference times for the models and added machine translation experiments. In the final version, we will also address and revise the format and references you mentioned.**
>
> >Q1: L120, 123: the quote for 'skipping' is wrongly formatted
>
> **Response**: Thank you for pointing out the formatting issue. We will correct the quotation marks for 'skipping' in lines 120 and 123 to ensure proper formatting.
>
> >Q2: L202-203 it should be either "The maximum Minimum Bayes Risk (MBR)..." or "The Minimum Bayes Risk (MBR)....", also it cites paper [20] (the Rouge package) which I think is incorrect for MBR.
>
> **Response**: Thank you for your reminder. We will correct this in the final version by citing the Diffusion-LM and DiffuSeq papers in this reference.
>
> >Q2: Besides the increased training time compared to DiffuSeq (L217-218), it would be beneficial to know the increased inference time too.
>
> **Response**: Thank you for this suggestion. We will supplement the increased inference time compared to DiffuSeq in the following Table.
>
> **Meta-DiffuB's computational complexity compared to DiffuSeq**
> |Method|increased parameters (%)|increased training time (%)|increased inference time (%)|
> |-|-|-|-|
> |Meta-Diffu$B$|2.2%|5%|0.5%|
>
> >Q3: How well does the Meta-Diffu$B$ model generalize to other types of text generation tasks beyond the Seq2Seq framework? Have experiments been conducted to evaluate its performance in different text generation scenarios?
>
> **Response**: We conducted extensive experiments to evaluate the generalization capabilities of the Meta-Diffu$B$ model across various text generation task. These tasks include generating informative dialogue responses (CC dataset), question generation (QT dataset), text simplification (WA dataset), and paraphrase generation (QQP dataset), as discussed in Section 4.1 and Appendix A. Thanks to the reviewers' suggestion, we supplemented additional machine translation datasets, as shown in the following table, and employed the same evaluation metrics used by SeqDiffuSeq and Dinoiser. Results where Meta-Diffu$B$ combined with different models show improved performance are indicated in bold. Our results show that Meta-Diffu$B$, when combined with different S2S-Diffusion models, achieves superior performance on machine translation tasks as well.
>
> **Experiment of Meta-Diffu$B$ on Machine Translation datasets**
> |Methods|SacreBLEU ↑ (IWSLT14 DE-EN)|SacreBLEU ↑ (WMT14 DE-EN)|
> |-|-|-|
> |DiffuSeq|29.43|22.72|
> |Meta-Diffu$B$ (exploiter=DiffuSeq)|**31.71**| **26.17**|
> |SeqDiffuSeq|30.16|23.28|
> |Meta-Diffu$B$ (exploiter=SeqDiffuSeq)|**32.41**| **26.14**|
> |Dinoiser|31.61|30.30|
> |Meta-Diffu$B$ (exploiter=Dinoiser)|**33.82**| **32.09**|
>
> >Q4: How does the model scale with larger datasets or more complex text generation tasks? Are there any specific optimizations or modifications required to handle real-world data volumes?
>
> **Response**: Meta-Diffu$B$ scales effectively with larger datasets, as demonstrated by our experiments on the Commonsense Conversation (CC) dataset with 3 million data points. We maintained consistent model parameters, architecture, and optimization methods across all datasets, including CC. Our framework's exceptional performance, even on large datasets, highlights its robustness and scalability. Additionally, as shown in the table from Q3, our Meta-Diffu$B$ performs well on machine translation tasks with the IWSLT14 dataset (160k data points) and the WMT14 dataset (4475k data points), without requiring any adjustments.

---

> > ### Comment · Reviewer_YnwD · 2024-08-12
> >
> > Thank you to the authors for their response. Since one of the key contributions of this paper is a strategy (noise scheduler) that enhances S2S-Diffusion models:
> > - I still have concerns about its generalizability, as the authors only consider continuous-based diffusion models and do not consider discrete-based models.
> > - While comparing S2S-Diffusion models before and after applying Meta-Diffu$\beta$ might be sufficient, the authors did not compare with other noise schedule baselines (as mentioned by reviewer qgkV).
> >
> > Given these points, I will maintain my score.

---

> > > ### Author Response · Authors · 2024-08-14
> > > **Thank you for your suggestion. To address your concerns, we have added more experiments.**
> > >
> > > >Q1: I still have concerns about its generalizability, as the authors only consider continuous-based diffusion models and do not consider discrete-based models.
> > >
> > > In the discrete S2S-Diffusion model, a set of $\beta$ values is still required to control the magnitude of the noise. Our scheduler inherently generates these $\beta$ values to regulate the noise. In the discrete S2S-Diffusion model, by multiplying the $\beta$ values generated by the scheduler with a Bernoulli random vector instead of Gaussian noise, our Meta-Diffu$B$ framework can be seamlessly integrated. As a result, we also include a comparison with the Reparameterized Diffusion Model (RDM) in the Table below.
> > >
> > > >Q2: While comparing S2S-Diffusion models before and after applying Meta-Diffu
> > >  might be sufficient, the authors did not compare with other noise schedule baselines (as mentioned by reviewer qgkV).
> > >
> > > In our study, we compared the noise scheduling from Denoiser, SeqDiffuSeq, and other S2S-Diffusion models as described in their papers. When paired with our scheduler, the noise scheduling consistently produced better results. The Diffusion-LEF paper (which proposed the Easy-First Schedule as noted by reviewer qgkV) used different datasets from ours. Therefore, we tested the Easy-First Schedule with DiffuSeq on the QQP and QG datasets. For a fair comparison, we used the Diffusion-LEF results without BERT. As shown in the table below, while the Easy-First Schedule improves DiffuSeq, it doesn't achieve the same level of enhancement as our Meta-Diffu$B$.
> > >
> > >
> > > **We compared more recent baselines with our Meta-Diffu$B$ on the QG and QQP datasets. These baselines include both discrete and continuous S2S-Diffusion models.**
> > > |Methods|BLEU ↑ (QQP)|BERTScore ↑ (QQP)|BLEU↑ (QG)| BERTScore ↑ (QG)|
> > > |-|-|-|-|-|
> > > |DiffuSeq|0.2413|0.8365|0.1731|0.6123|
> > > |Diffusion (Easy-First Schedule)|0.2503|0.8692|0.1812|0.6253|
> > > |Meta-Diffu$B$ (exploiter=DiffuSeq)|**0.2552**| **0.8821**|**0.1826**|**0.6357**|
> > > |DiffuSeq-v2|0.2411|0.8393|-|-|
> > > |Meta-Diffu$B$ (exploiter=DiffuSeq-v2)|**0.2556**| **0.8829**|-|-|
> > > |BG-DiffuSeq|0.2619|0.8427|0.1744|0.6280|
> > > |Meta-Diffu$B$ (exploiter=BG-DiffuSeq)|**0.2790**| **0.8757**|**0.1838**|**0.6571**|
> > > |TESS|0.3020|0.8570|0.1950|0.6580|
> > > |Meta-Diffu$B$ (exploiter=TESS)|**0.3142**|**0.8975**| **0.2055**|**0.6761**|
> > > |RDM (Discrete Diffusion)|0.2510|0.8472|0.1802|0.6310|
> > > |Meta-Diffu$B$ (exploiter=RDM)|**0.2684**|**0.8724**|**0.2271**|**0.6542**|

---

### Official Review · Reviewer_SoCC · 2024-07-13

**Soundness:** 3
**Presentation:** 4
**Contribution:** 3
**Rating:** 6
**Confidence:** 4

**Summary:**

This paper proposes a meta-learning based noise scheduler to incoporate contextualisation in texts. The scheduler is a plug-and-play module which could be applied to other similar sequential setting. The experimental results demonstrate its superiority comparing to the baselines, in terms of generating quality and diversity. Besides, the ablation study investigates into and provide insights about the relationship of the difficulty levels and noise schedule, which is quite interesting. The paper is well organised and easy to follow.

**Strengths:**

It is the first work to take the contextilisation into consideration when doing adaptive noise schedule for text generation. Its effectiveness has been demonstrated, upon the questions listed below are addressed.

The experiments are well planned to support the arguments proposed. The presentation is clear and easy to follow.

**Weaknesses:**

While the diffusion process is now designed to incoporate a time variable noise schedule, the denoising process is still supposed to handle fixed denoising steps. More insights on how this will be handled effectively would be expected.

Some statements in the text seems not align well with the information disclosing by the figures. For example, Figure 3, the noise level is quite stable for the proposed method but in the text below, it claims 'Meta-DiffuB applies adaptive noise at varying training epchos'. I am wondering if there are any typos involved. Another example is in Figure 4 at least for QQP dataset - Meta-DiffuB applies less noise than the other methods as its curves lay below the curves of the other methods. thus, the claim 'This noise-scheduling approach—applying more noise to the harder sentences (H) and less to the easier sentences' looks conflict with such observation from the figure.

Besides, as algorithm 1 alternates the training of scheduler and exploiter, It would be guideful if some insights about the convergence rate are provided.

**Questions:**

I have three major concerns as listed in the weakness section about the reasonability, the consistence of figures and in-text description about some experiemtnal results, and the time complexity of training algorithm.

**Limitations:**

I suspect the main limitation would be the converge speed but it was not mentioned in the paper.

The schedular is implemented with a Seq2Seq structure, and other options could be provided as well to inspire other potential application scenarios.

---

> ### Author Response · Authors · 2024-08-07
> **Thank you for your valuable suggestions. We have included the training curves and addressed your comments as below.**
>
> >Q1: For example, Figure 3, the noise level is quite stable for the proposed method but in the text below, it claims 'Meta-Diffu$B$ applies adaptive noise at varying training epchos.
>
> **Response**:
> In Figure 3, our method differs from approaches like SeqDiffuSeq or Dinoiser, which rely on predefined rules. Instead, our approach is data-driven, with the model learning the underlying patterns. As a result, the degree of variation—whether large or small—is not the primary focus. Additionally, the learned noise can be flexibly integrated into other systems in a plug-and-play manner.
>
> >Q2: Another example is in Figure 4 at least for QQP dataset - Meta-Diffu$B$ applies less noise than the other methods as its curves lay below the curves of the other methods. thus, the claim 'This noise-scheduling approach—applying more noise to the harder sentences (H) and less to the easier sentences' looks conflict with such observation from the figure.
>
> **Response**: We appreciate the opportunity to correct our previous statement. We will correct this error in the final version.
>
> >Q3: Besides, as algorithm 1 alternates the training of scheduler and exploiter, It would be guideful if some insights about the convergence rate are provided.
>
> **Response**: We supplement the [Training Curve Image](https://github.com/metabeta-diffusion/metabeta-diffusion/blob/main/img/training%20curve.jpg?raw=true). Our Meta-Diffu$B$ shows Our model converges significantly faster than DiffuSeq, achieving convergence within just 10k epochs.

---

### Author Response · Authors · 2024-08-07
**General Response**

We sincerely thank all the reviewers for their valuable suggestions. We have addressed all the comments as below.

Our contributions include:
1. **General Applicability:** Meta-Diffu$B$ can be integrated with various S2S-Diffusion models, enhancing their performance through our noise scheduling method. Section 4.5 demonstrates integration with DiffuSeq, while Appendix F shows experiments with SeqDiffuSeq and Dinoiser, confirming our method's broad applicability and significant performance improvement.
2. **Efficiency:** We used the simple LSTM-Seq2Seq model as our scheduler for all tasks. As shown in Table 1 (Section 4.5), even with this small parameter scheduler combined with DiffuSeq, we achieved the best performance across four datasets.
3. **Task Performance:** Our model consistently outperforms previous models in tasks such as Commonsense Conversation (CC), Quasar-T (QT), Wiki-Auto (WA), and Quora Question Pairs (QQP) with our innovative noise scheduling approach.
4. **Versatility:** 1. Training from Scratch: As shown in Table 1 (Section 4.5) and Table 4 (Appendix F), Meta-Diffu$B$ achieves superior results when training our scheduler with DiffuSeq and other S2S-Diffusion models from scratch. 2. Plug-and-Play: In Tables 3 and 8 (Sections 4.8 and Appendix I), we demonstrate that our scheduler, trained on different datasets, can be seamlessly integrated with various S2S-Diffusion models without any fine-tuning.

In this rebuttal, We supplement our work with training curve, machine translation experiments, plug-and-play tests with other S2S-Diffusion models, and the inclusion of more recent baseline models. These experiments will be incorporated in the latest version.

**Training curve**: [Training Curve Image](https://github.com/metabeta-diffusion/metabeta-diffusion/blob/main/img/training%20curve.jpg?raw=true)

**Meta-Diffu$B$'s computational complexity compared to DiffuSeq**
|Method|increased parameters (%)|increased training time (%)|increased inference time (%)|
|-|-|-|-|
|Meta-Diffu$B$|2.2%|5%|0.5%|

**Experiment of Meta-Diffu$B$ on Machine Translation dataset**
|Methods|SacreBLEU ↑ (IWSLT14 DE-EN)|SacreBLEU ↑ (WMT14 DE-EN)|
|-|-|-|
|DiffuSeq|29.43|22.72|
|Meta-Diffu$B$ (exploiter=DiffuSeq)|**31.71**| **26.17**|
|SeqDiffuSeq|30.16|23.28|
|Meta-Diffu$B$ (exploiter=SeqDiffuSeq)|**32.41**| **26.14**|
|Dinoiser|31.61|30.30|
|Meta-Diffu$B$ (exploiter=Dinoiser)|**33.82**|**32.09**|

**We compared more recent baselines with our Meta-Diffu$B$ on the QG and QQP datasets. These baselines include both discrete and continuous S2S-Diffusion models.**
|Methods|BLEU ↑ (QQP)|BERTScore ↑ (QQP)|BLEU↑ (QG)| BERTScore ↑ (QG)|
|-|-|-|-|-|
|DiffuSeq|0.2413|0.8365|0.1731|0.6123|
|Diffusion (Easy-First Schedule)|0.2503|0.8692|0.1812|0.6253|
|Meta-Diffu$B$ (exploiter=DiffuSeq)|**0.2552**| **0.8821**|**0.1826**|**0.6357**|
|DiffuSeq-v2|0.2411|0.8393|-|-|
|Meta-Diffu$B$ (exploiter=DiffuSeq-v2)|**0.2556**| **0.8829**|-|-|
|BG-DiffuSeq|0.2619|0.8427|0.1744|0.6280|
|Meta-Diffu$B$ (exploiter=BG-DiffuSeq)|**0.2790**| **0.8757**|**0.1838**|**0.6571**|
|TESS|0.3020|0.8570|0.1950|0.6580|
|Meta-Diffu$B$ (exploiter=TESS)|**0.3142**|**0.8975**| **0.2055**|**0.6761**|
|RDM (Discrete Diffusion)|0.2510|0.8472|0.1802|0.6310|
|Meta-Diffu$B$ (exploiter=RDM)|**0.2684**|**0.8724**|**0.2271**|**0.6542**|


**Plug-and-play experiments on SeqDiffuSeq integrated with our scheduler**
|Scheduler|SeqDiffuSeq|BLEU ↑|BERTScore ↑|Dist-1 ↑|
|-|-|-|-|-|
|WA|QQP|**0.2627**|**0.8481**|**0.9814**|
|Null|QQP|0.2434|0.8400|0.9807|
|WA|QT|**0.1834**|**0.6226**|**0.9369**|
|Null|QT|0.1746|0.6174|0.9248|

**Plug-and-play experiments on Dinoiser integrated with our scheduler**
|Scheduler|Dinoiser|BLEU ↑|BERTScore ↑|Dist-1 ↑|
|-|-|-|-|-|
|WA|QQP|**0.2079**|**0.8121**|**0.9765**|
|Null|QQP|0.1949|0.8036|0.9723|
|WA|QT|**0.0495**|**0.4740**|**0.8289**|
|Null|QT|0.0477|0.4690|0.8191|

---

### Decision · Program_Chairs · 2024-09-25

**Decision:**

Accept (poster)

**Comment:**

This paper presents a novel approach for contextualized noise scheduling in sequence-to-sequence diffusion models. While initially there were some concerns on the novelity side and the up-to-date experiments, I think the authors have addressed these well in their rebuttal. They provided additional experiments comparing to recent baselines including both new continous diffusion baselines and discrete diffusion models, demonstrated applicability to machine translation tasks, and showed plug-and-play capabilities with other models. The core idea is innovative and results show consistent improvements over existing methods across multiple datasets and tasks. Given the thorough response and additional results, I recommend accepting this paper despite its borderline status.